# Improving the Quality and Safety of Pu-erh Tea by Inoculation of *Saccharomyces cerevisiae* and *Lactobacillus plantarum*

Yali Wang [1,2,†], Yifei Liu [2,†], Zhaoyue Sun [2], Joseph Brake [3], Yuhuan Qin [2], Jing Li [2,*] and Xiaobin Wu [2,*]

1 Department of Chemistry and Chemical Engineering, Yulin University, Yulin 719000, China
2 Development Center of Plant Germplasm Resources, College of Life Sciences, Shanghai Normal University, Shanghai 200234, China
3 Department of Biochemistry and Redox Biology Center, University of Nebraska-Lincoln, Lincoln, NE 68588, USA
* Correspondence: lijing52@shnu.edu.cn (J.L.); xwu6@shnu.edu.cn (X.W.); Tel.: +86-21-64322762 (X.W.)
† These authors contributed equally to this work.

**Abstract:** The sensory quality and health benefits of Pu-erh tea are mainly determined by microbial fermentation processing. The directed exogenous inoculation of specific microorganisms is an effective method to improve the quality and flavor of Pu-erh tea. In this study, *Lactobacillus plantarum* and *Saccharomyces cerevisiae* were introduced into the fermentation processes of Pu-erh tea, as they are the main contributors to enzyme secretion, to change the tea's functional components. The raw tea materials, spontaneous fermentation tea and microbiological fermentation tea were analyzed by microbiomics and metabolomics. A total of 248 metabolites were characterized, 71 of which were identified as essential metabolites involved in the metabolic changes. These essential metabolites were produced by specific dominant microbial species with multivariate analysis methods. Metabolites essential to the sensory quality and health benefits of Pu-erh tea, such as flavonoids and free amino acids, were increased in tea samples inoculated with *Lactobacillus plantarum* and *Saccharomyces cerevisiae* following fermentation. Fungal diversity decreased after fermentation, and both the diversity and richness of bacteria were significantly decreased. In conclusion, our results demonstrate the advantages of *Lactobacillus plantarum* and *Saccharomyces cerevisiae* in forming the unique sensory characteristics of Pu-erh tea, and they indicate that the microbial composition is a key factor in altering the tea's metabolic profile. Our work establishes a theoretical foundation for the promotion of the safety and quality of Pu-erh tea through exogenous inoculation with *Lactobacillus plantarum* and *Saccharomyces cerevisiae*.

**Keywords:** Pu-erh tea; yeast; microbiome; metabolomics; processing technology

## 1. Introduction

Pu-erh tea (PET) is a well-known fermented dark tea exclusively produced by the spontaneous fermentation of *Camellia assamica* tea leaves [1]. During the fermentation process, a series of chemical reactions are involved, such as decomposition, oxydoredution, polymerzation, structural modification, methylation and glycosylation. The sensory quality and multiple health benefits of PET, including anti-diabetic [2], anti-oxidative [3], anti-cancerogenic [4], anti-bacterial [5], anti-inflammatory [6] and free radical scavenging effects, are largely determined by microbial fermentation. Previous studies have reported that the chemicals of PET are hydrolyzed and conversed by microbial metabolites in hot and humid conditions [7]. For instance, the content of catechin derivatives, flavonoids and their glycosides, phenolic acids, alkaloids and terpenoids is significantly changed during PET fermentation [8]. The critical element involved in generating the bioactivity, quality and sensory characteristics of Pu-erh tea are interrelated with complex biochemical changes, which result from the microorganisms present and the metabolites during the fermentation process.

The natural solid-state fermentation (SSF) process is usually used for the manufacture of Pu-erh tea. In this process, sun-dried green tea leaves are moistened with water and fermented for a few weeks [9]. Recently, both researchers and the agricultural industry have utilized microbial fermentation to increase the nutritional value, flavor and aroma of food. These products have been applied as feed additives to promote the performance of livestock [10]. Traditional Pu-erh tea is fermented spontaneously using raw materials in hot and humid conditions without a seeding starter strain. The spontaneous fermentation process changes the microbial diversity and results in a rise in the abundance of certain microorganisms, such as *Aspergillus niger*, *Rasamsonia emersonii* and *Thermomyces lanuginosus* [11]. *L. plantarum* are generally recognized as safe and effective microorganisms that have been utilized in the processing of fermenting food for centuries. Previous studies showed that *L. plantarum* strains isolated and identified from tea leaves could inhibit the growth of *Salmonella typhi*, *Escherichia coli*, *Staphylococcus aureus*, *Enterococcus faecalis* and *Citrobacter* sp. [12]. Furthermore, *L. plantarum* produces extracellular tannase, which is beneficial for health. Both culture and culture-independent studies showed that *L. plantarum* changes the tea leaf components, such as organic acids, free amino acids and catechins, during fermentation [13].

*S. cerevisiae*, a unicellular fungus, is of great importance for various biotechnological applications relating to its fermentation ability, accompanied by the production of $CO_2$ and alcohol, and its tolerance to unfavorable conditions of osmolarity and low pH. The most prominent application involving the use of *S. cerevisiae* is in food fermentation. *S. cerevisiae* is a valuable tool for the fermentation process due to its "make–accumulate–consume" lifestyle [14]. This results from the Crabtree effect, which consists of the fact that *S. cerevisiae* does not use respiratory machinery to metabolize saccharides and instead produces ethanol and $CO_2$, even under aerobic conditions. *S. cerevisiae* is the dominant sugar fermenter due to its remarkable resistance to high sugar levels and its production of different aromatic, volatile compounds. The function of *S. cerevisiae* in beverage [15], bread [15,16] and biofuel production [17] has been characterized and explored. However, the roles of *S. cerevisiae* in tea fermentation have not been investigated.

Both *L. plantarum* and *S. cerevisiae* are applied widely in food fermentation and the large-scope production of organic acids, enzymes and bioactive compounds. They produce many enzymes efficiently and these enzymes have been used commercially. Well-combined microorganism diversity during the fermentation process has the ability to control the metabolic outcome and therefore product quality. As such, it has been considered to combine *L. plantarum* and *S. cerevisiae* as a tea fermentation starter strain to give rise to the growth of diverse microorganisms and convert the tea compounds involved in the formation of the characteristic bioactive function. However, the relationships between the metabolites that characterize Pu-erh tea and a fermentation microbiome mainly composed of *L. plantarum* and *S. cerevisiae* have not been clarified. It is necessary to understand these relationships for the purpose of enhancing the quality of Pu-erh tea.

In this study, raw material tea, spontaneous fermentation tea and microbial fermentation tea were analyzed to investigate microbiome composition changes and the correlation between the microbiome and metabolites. The hundreds of endogenous metabolites from the Pu-erh tea fermentation process were characterized by the powerful metabolomics method combined with biochemical measurements. Multivariate analysis combined with microbiomics (high-throughput sequencing combined with qPCR) was applied to explore the metabolic potential in the microbial community. Moreover, the changes in functional compounds in Pu-erh tea during the manufacturing process were discussed. Furthermore, the intricate relationship between the effective microorganisms, metabolic pathways and dominant functional compounds in Pu-erh tea was evaluated. *L. plantarum* in conjunction with *S. cerevisiae* as the starter strain applied in Pu-erh tea fermentation exhibited the potential to improve the tea quality.

## 2. Materials and Methods

### 2.1. Preparation of Tea Leaves

Pu-erh tea raw materials (sun-dried green tea leaves) were obtained from the Yunnan LongRun Tea Industry Group (Yunnan, Kunming, China). A haploid control *S. cerevisiae* strain (BY4741) was obtained from Open Biosystems. Synthetic complete (SC) medium with 1.5% agar was added to culture *S. cerevisiae* in plates at 30 °C. *L. plantarum* (1.16089) were donated by the Institute of Microbiology, Chinese Academy of Sciences.

The Pu-erh tea raw materials (100 g) were mixed with distilled water (30 mL) to produce spontaneous fermentation tea (SFT). Li-mupirocin-modified MRS medium containing *L. plantarum* and *S. cerevisiae* (10 mL, $1 \times 10^8$ Cells/mL) was sprayed on sun-dried tea leaves (100 g) and fermented at 37 °C for 3 weeks; these samples were named microbial fermentation tea (MFT).

### 2.2. Chemicals

The standard reagents involved in tea metabolites were purchased from the Yuanye Biotechnology Company (Shanghai, China). Acetonitrile, formic acid and methanol of LC–MS grade were obtained from Sigma Aldrich (St. Louis, MO, USA). Other reagents ($\geq$98%) were obtained from the China National Medicines Corporation Ltd. (Beijing, China).

### 2.3. Gene Sequencing and Quantitative PCR (qPCR) Detection

The FastDNA® Spin Kit (MP Biomedicals, Norcross, GA, USA) was used to extract microbial DNA from tea samples according to the instructions. The extracted 16S rRNA gene in the V3-V4 region was amplified using forward primer 338F (5′-ACTCCTACGGGAGGCAGCA-3′) [18] and reverse primer 806R (5′-GGACTACHVGGGT WTCTAAT-3′) [19]. Before amplifying the DNA in triplicate, the sample was subjected to electrophoresis on 2% agarose gels. A total volume of 20 μL PCR products were purified with the AxyPrep DNA Gel Extraction Kit (Axygen Biosciences, Union City, CA, USA). The QuantiFluotTM DNA Assay Kit (Pro-Mega, Madison, WI, USA) was applied for the quantification of the purified PCR products. Following quantification, the Illumina MiSeq system from Majorbio Bio-Pharm Technology Co. (Shanghai, China) was employed to identify the mixed sample amplicon sequencing.

The raw sequencing data were demultiplexed using Trimmomatic (v1.7.0) for denoising and trimming [20], FLASH for quality filtering [21], UPARSE for pairing [22] and UCHIME for alignment [23]. The UPARSE software (version 7.1 http://drive5.com/uparse/ accessed on 27 March 2022) clustered the resulting sequences into operational taxonomic units (OTUs) with a 97% similarity threshold. The RDP classifier (http://rdp.cme.msu.edu/, accessed on 27 March 2022) assigned the OTUs from representative sequences in taxonomic information with a 70% threshold by comparison with the Silva database (SSU128). The Mothur software (version.1.47.0) developed by Dr. Patrick Schloss group (Michigan State University, Microbiological Sciences and Immunology, East Lansing, MI, USA), which was utilized to estimate the community richness index (CHAO index), community diversity (Shannon index) and the Good's coverage of sequencing.

The ITS gene of *S. cerevisiae* in a 188 bp fragment (091-279) was cloned using PCR. In the p416-TEF vector, specific primers (F: 5′-CGCGGATCCCCAGCCG GGCCTGCGCTTAAG, R: 5′- CCGCTCGAGCCTCTGGGCCCCGA TTGCTCG) were inserted into the BamHI and XhoI sites to produce the p416-TEF-ITS (188) plasmid for fungal standard curve generation. Meanwhile, the specific primers (F: 5′-CGCGGAT CCCGGCAGGCCTAACACATGCAAG, R: 5′-CCGCTCGAGGCATTTCACCG CTACACCTG) from a 659 bp fragment (031-690) of the E. coli 16s rDNA was inserted into the BamHI and XhoI sites in the p416-TEF vector. After confirming the gene sequences of all plasmids, the diluted plasmid DNA was used to generate standard curves for quantitative PCR (qTOWER 3.0G) [24]. The molecular biology techniques followed previously established methods [25]. Following quantitative PCR, TIANamp Soil DNA kit (Tiangen Biotech, Beijing, China) was used to extract DNA from

the microorganisms for genomic analysis. The method of extracting microorganisms from different Pu-erh tea samples was based on our previous study [11].

### 2.4. Metabolomics Analysis

First, 0.1 g tea powder was extracted using 3 mL 70% methanol for 30 min in an ultrasonic bath at 60 °C. The filtered extract liquid was subjected to metabolomics analysis using ultra-performance liquid chromatography coupled with electrospray time-of-flight mass spectrometry (UPLC-Q-TOF/MS) (waters, Milford, CT, USA). The chromatography separations were performed on a BEH C18 column (100 mm × 2.1 mm, 1.7 μm; Waters corporation, Milford, MA, USA). The column was eluted with water as mobile phase A (0.1% formic acid) and acetonitrile as mobile phase B (0.1% formic acid). The gradient sequence was as follows: 0–3 min, 5% B; 3–10 min, 20% B; 10–15 min, 100% B. The flow rate was 0.4 mL/min, column temperature 45 °C, injection volume 5 μL. The Q-TOF mass spectrometer was equipped with an electrospray ionization (ESI) source operating in negative ion mode, and the data were collected from 50 to 1000 m/s. The de-solvation temperature was 450 °C, and the source temperature was 115 °C, with a cone gas flow of 15 L/min. The capillary voltage, sampling cone voltage and collision energy were 2000 V, 40 V and 6 eV, respectively. Three independent extractions and analyses were performed.

The tea polyphenols, total amino acids, tea proteins and water-soluble sugars were extracted and quantitated using the method of Li et al. [11]. The raw data obtained from LC–MS were initially processed using the MetaboAnalyst software (https://www.metaboanalyst.ca/, accessed on 28 March 2022). The SIMCA-P software (version 14.1, Umetrics AB, Umea, Sweden) was applied to process the acquired data and evaluate the metabolite changes in different tea samples.

Hierarchical cluster analysis (HCA) was used to plot the respective dendrograms and their relationships. Unsupervised principal component analysis (PCA) distinguished the differences among the samples through the intrinsic variation from the collected data matrix. Meanwhile, supervised orthonormal partial least-squares discriminant analysis (OPLS-DA) was applied to classify samples of solely Y variables, and it was combined with a threshold of variable importance projection (VIP) > 1.16 to identify the critical metabolites that caused the metabolomic variations during the manufacturing process. The data were analyzed and a heat map was drawn through the website (https://software.broadinstitute.org/morpheus/, accessed on 28 March 2022).

### 2.5. Statistical Analysis

All experiments were carried out in triplicate and the results were expressed as mean values followed by the standard deviation ($n$ = 3). Differences in the relative abundance of operational taxonomic units (OTUs) were assessed using the Bray–Curtis distance, which was calculated using ANOSIM/Adonis dissimilarity analyses. The significance level of Pu-erh tea metabolites between different groups was calculated by one-way analysis of variance (ANOVA) with Dunnett's multiple comparisons test using the GraphPad Prism 7.00 software (GraphPad Software Inc., San Diego, CA, USA). To estimate the community richness index (CHAO index) and community diversity (Shannon index), the statistical T-test method was used to determine the significant differences between samples. The linear correlation between the effects of microbes and the chemical composition of the tea samples was analyzed using the SMICA-P software (version 14.1, Umetrics AB, Umea, Sweden) and the probability values. $p$-values below 0.05 were considered significant.

### 3. Results

### 3.1. Sequence Statistics of Microorganisms in Tea Samples

To comprehensively investigate the microbiomes of different fermentation tea samples (Pu-erh tea raw materials, spontaneous fermentation tea and microbial fermentation tea), PCR with specific primers was used to clone specific regions of the fungal ITS gene and bacterial 16s rDNA. High-throughput sequencing (Illumina sequencing) was applied to

sequence the PCR fragments. A total of 564,007 qualified fungal raw sequences (Table 1) and 402,036 qualified bacterial sequences (Table 2) were obtained from the Pu-erh tea samples. The total sequence lengths of fungi and bacteria were 142,413,586 bp and 170,484,585 bp, respectively. The average sequence length of bacteria was 424 bp, which was higher than the average fungal sequence length of 252 bp.

**Table 1.** The optimized sequence analysis based on fungal high-throughput sequencing.

| Amplified Region | Sample | Sequences | Bases (bp) | Average Length |
|---|---|---|---|---|
| ITS1F_ITS2R | 9 | 564,007 | 142,413,586 | 252 |

**Table 2.** The optimized sequence analysis based on bacterial high-throughput sequencing.

| Amplified Region | Sample | Sequences | Bases (bp) | Average Length |
|---|---|---|---|---|
| 338F_806R | 9 | 402,036 | 170,484,585 | 424 |

### 3.2. Overview of Microbial Community and Richness

For fungi and bacteria, as the most abundant microorganisms in Pu-erh tea, the community and richness are related to the quality and safety of the tea products. From the quality-filtered fungal sequence results, the three tea samples were clustered into 568 fungal operational taxonomic units (OTUs) at a 97% similarity cut-off. We used Venn diagram analysis to show the shared OTUs of fungi in different tea samples. A total of 87 fungal OTUs were shared in the three Pu-erh tea samples, and the highest fungal OTU number of 384 appeared in the PET sample (Figure 1A), which indicated that the number of fungal OTUs decreased after the fermentation process. Moreover, the Chao index suggested that the MFT sample had the highest fungal richness compared to the PET and SFT samples (Figure 1B). The Shannon index indicated that the PET sample exhibited the highest fungal diversity (Figure 1C). The fungal rarefaction curves fully approached the saturation plateau (Figure S1A). Furthermore, these results collectively demonstrated that the fungal diversity was significantly decreased in the SFT sample.

On the other hand, the fungal taxonomies were identified across all samples. At the species level, unclassified *Aspergillus* and *A. penicillioides* comprised > 80% of all sequences in the PET sample. *Aspergillus* was the predominant genus in the SFT sample, accounting for 99% of the total effective sequences. *S. cerevisiae* was the most abundant fungal species in the MFT sample (Figure 1D). *Aspergillus* was dominant in all Pu-erh tea samples (Figure 1E). *Aspergillus* is known to contribute to the digestion of macromolecules (polysaccharides, proteins and lipids) into small molecules to increase the nutritional profile or change the tea flavor during Pu-erh tea manufacturing.

To analyze the bacterial microbiome in Pu-erh tea, the quality-filtered bacterial sequences of the three Pu-erh tea samples were clustered into 684 bacterial OTUs at a 97% similarity cut-off. The Venn diagram analysis showed that the three different manufactured tea samples shared 93 bacterial OTUs, the total OTU numbers of bacteria were decreased after the fermentation processes, and the lowest OTU number of 156 appeared in the MFT sample (Figure 2A). From Figure 2B, the Chao index result indicated that the lowest bacterial richness was observed in the MFT sample. Similar results were displayed for the Shannon index, suggesting that the MFT sample displayed the lowest bacterial diversity (Figure 2C). The bacterial rarefaction curves fully approached the saturation plateau (Figure S1B).

Contrary to the fungal community, the bacterial richness and diversity in the SFT sample were higher than in the MFT. The predominant bacterial genus differed across the three samples (Figure 2D). At the genus level, unclassified *Pedobacter* was the dominant bacterium in the raw Pu-erh samples. *Klebsiella variicola* and *unclassified Enterobacterales* were increased in the SFT sample. *L. plantarum* was the most abundant in the MFT sample. The norank *Chloroplast* was the obvious dominant bacteria in the raw Pu-erh tea sample, but it sharply decreased after fermentation, and, instead, unclassified *Enterobacterales*' abundance

increased to 23% in the SFT sample. *L. plantarum* dramatically reproduced to 98% in the MFT sample (Figure 2E). The differences in the microbial community structures in different tea samples were assessed using the Bray–Curtis distance. The sample distance distribution was displayed by ANOSIM/Adonis dissimilarity analyses. The results suggested that the sample distance distribution within groups was smaller than between groups and all the experimental data had statistical significance (Figure S2A,B).

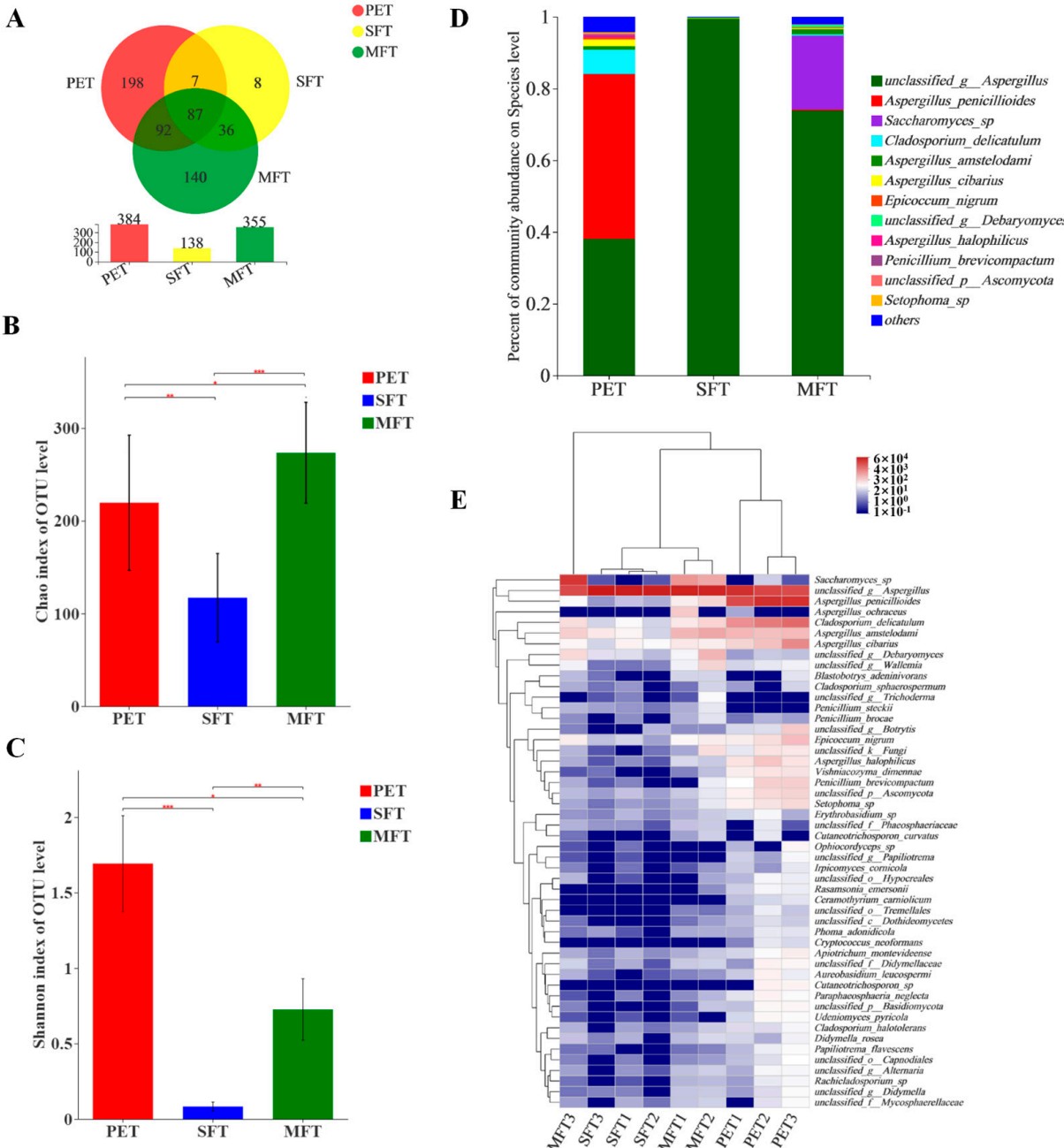

**Figure 1.** Analysis of fungal richness and composition in different processed tea samples. (**A**) Venn diagram of fungal OTU level in different processed Pu-erh tea samples. (**B**) Chao index and (**C**) Shannon index. (**D**) Fungal taxonomic compositions displaying the fungal successions at the species level. The fungal communities in the three samples are comparably diverse. (**E**) Heat map comparisons of fungal communities in Pu-erh tea samples at the species level. (Note: The taxonomic abundance < 1% was classified as "others"; "*" $p < 0.05$; "**" $p < 0.01$; "***" $p < 0.001$.)

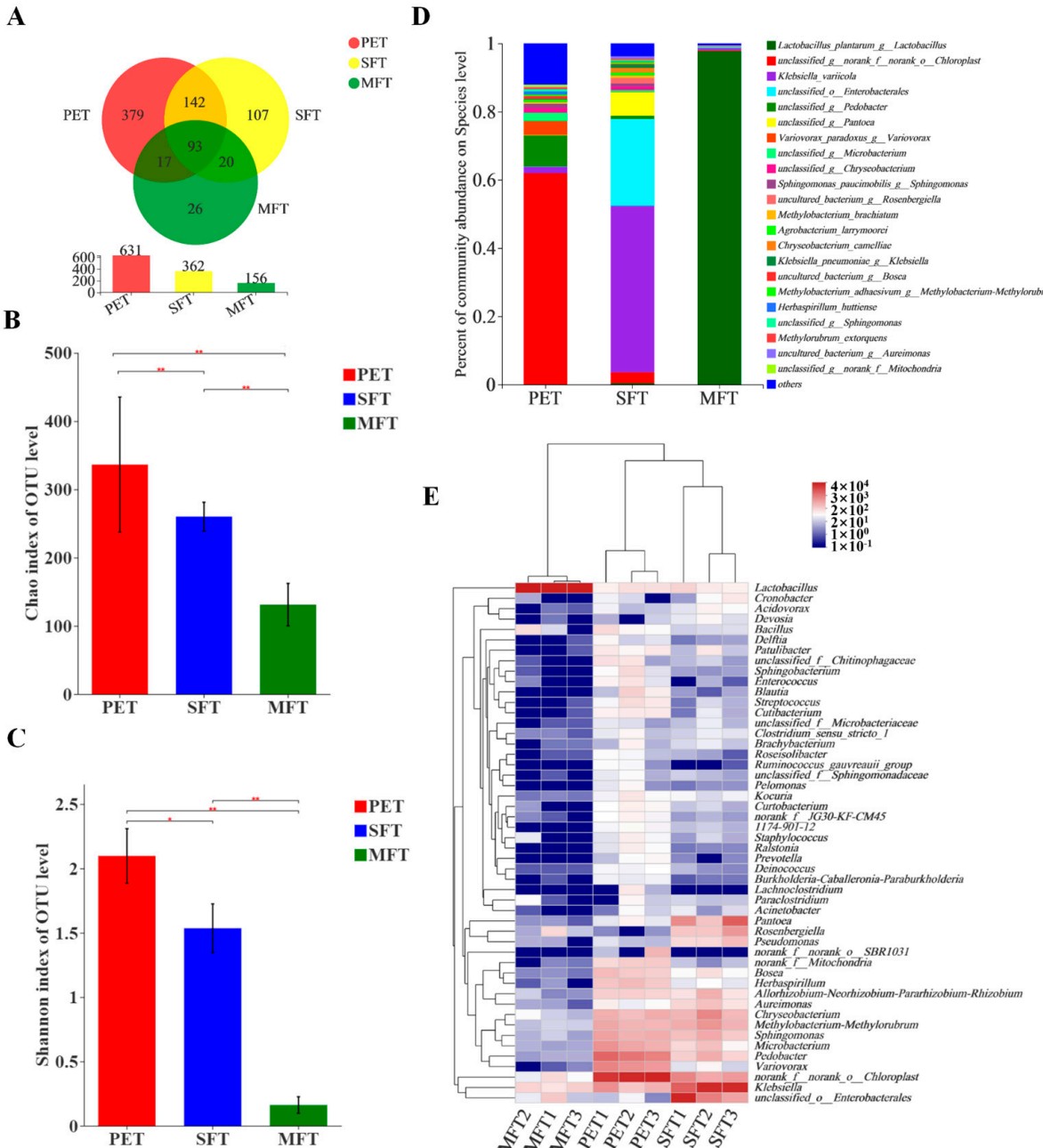

**Figure 2.** Analysis of bacterial richness and composition in different processed tea samples. (**A**) Venn diagram of bacterial OTU level in different processed Pu-erh tea samples. (**B**) Chao index and (**C**) Shannon index. (**D**) Bacterial taxonomic compositions displaying the bacterial successions at the genus level. The bacterial communities in the three samples are comparably diverse. (**E**) Heat map analysis of bacterial communities in Pu-erh tea processed samples at the species level. (Note: The taxonomic abundance < 1% was classified as "others"; "*" $p < 0.05$; "**" $p < 0.01$).

### 3.3. Overview of the Relationship between Metabolism and Microorganisms

The quality and bioactivity of Pu-erh tea is mainly determined by its biochemical composition, which is affected by the types and quantities of microorganisms. In order to understand the connection between microorganisms and biochemical changes, the metabolic pathways of microorganisms were studied and predicted in this work. Based on the typing analysis, the tea samples could be clustered into two groups at the OTU level. The PET sample was significantly distinguished from the MFT and SFT samples (Figure 3A). Fungal flora were classified into eight types, the community structure was

shifted into nine types, and the community structures were changed after the fermentation process (Figure 3A). The PCA score plot results suggested that the tea samples were divided into three successive processes, and nine bacterial types were identified by the total genus numbers (Figure 3B). The results revealed that the bacterial community structures were significantly different between the microbial and spontaneous fermentation samples.

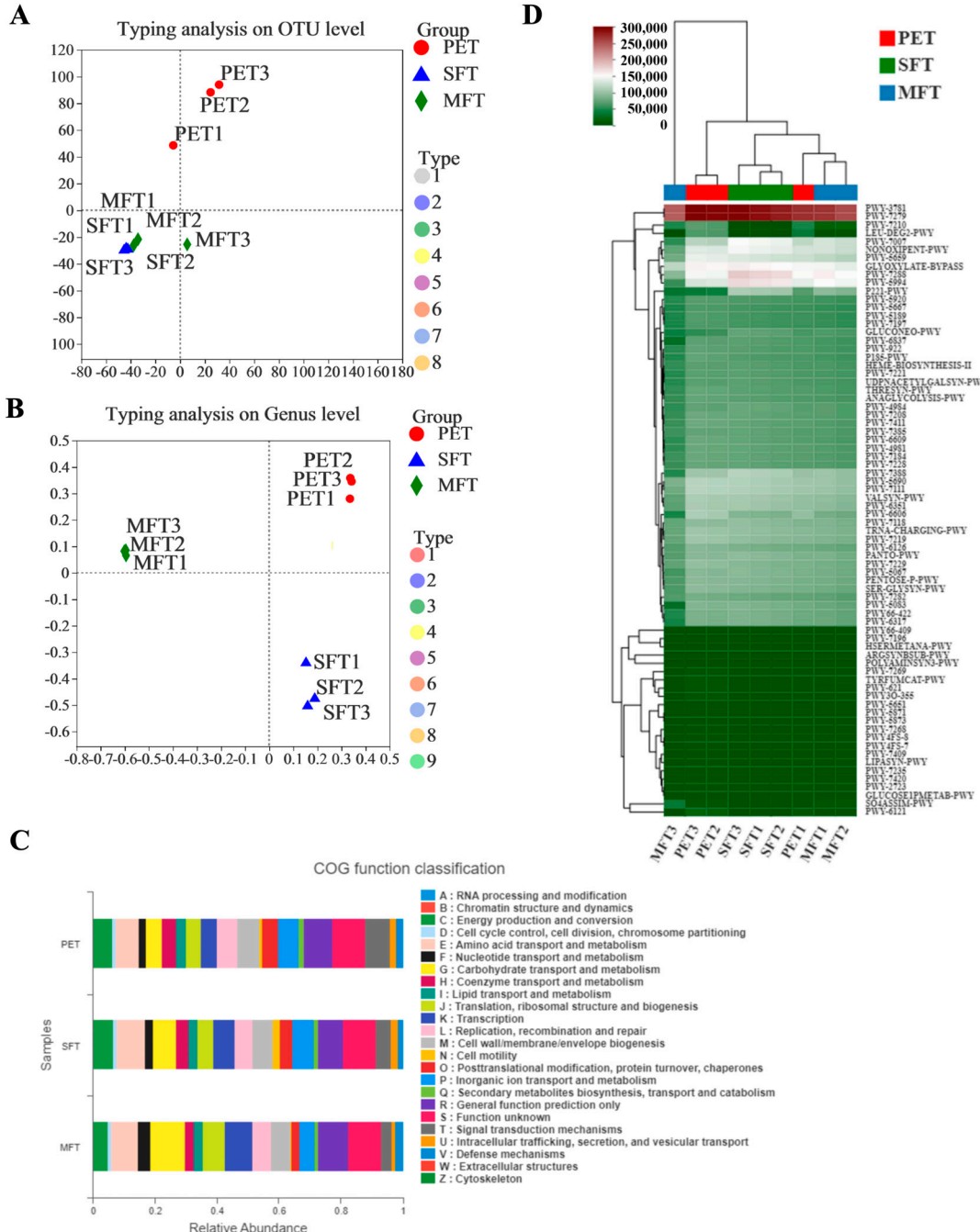

**Figure 3.** Predicted metabolic pathways according to microbial compositions in Pu-erh tea processed samples. Classification of (**A**) fungal and (**B**) bacterial flora through OTU level. (**C**) Statistical chart of bacteria participating in metabolic activities. (**D**) Heat map of fungi involved in metabolic pathway.

The metabolic activities that correlated to the fungal communities were predicted effectively by PICRUSt2 (http://huttenhower.sph.harvard.edu/galaxy, accessed on 28 March 2022) function prediction [26], and the KEGG database was used for description of the metabolic pathways (Table S1). The correlation between metabolic activities and the Pu-erh tea sam-

ples was manifested by the color depth (Figure 3D). On the other hand, the Cluster of Orthologous Genes (COG) database was used to classify the microbial gene functions, and the different bacterial pathways predicted by KEGG are shown in Figure 3C. Within the metabolism category, the relative abundance of pathways associated with carbohydrate transport and metabolism increased in MFT. It indicated that fungal metabolism contributed to the hydrolysis of macromolecular compounds, while bacterial metabolism was mainly involved in the decomposition, transformation and transfer of small molecular compounds in the fermentation process.

To investigate the metabolome changes after different fermentation processes, UPLC–MS-based metabolomics analysis coupled with multivariate analysis was applied to determine the critical metabolites that responded with metabolome variations during fermentation. A total of 248 metabolites were identified in three samples (Table S2). A Venn diagram was constructed and showed that 154 metabolites were identified in all tea samples (Figure 4A). Among the remaining 94 metabolites, 93 metabolites were from the MFT sample and one metabolite from the PET sample. Moreover, 89 metabolites were specific to fermentation tea samples. These results showed that metabolite transformation happened during microbial fermentation processing. The total ion chromatogram of each Pu-erh tea sample is exhibited in Figure 4B. The identification of metabolites in tea samples depended on authentic standards and a tea metabolomics database.

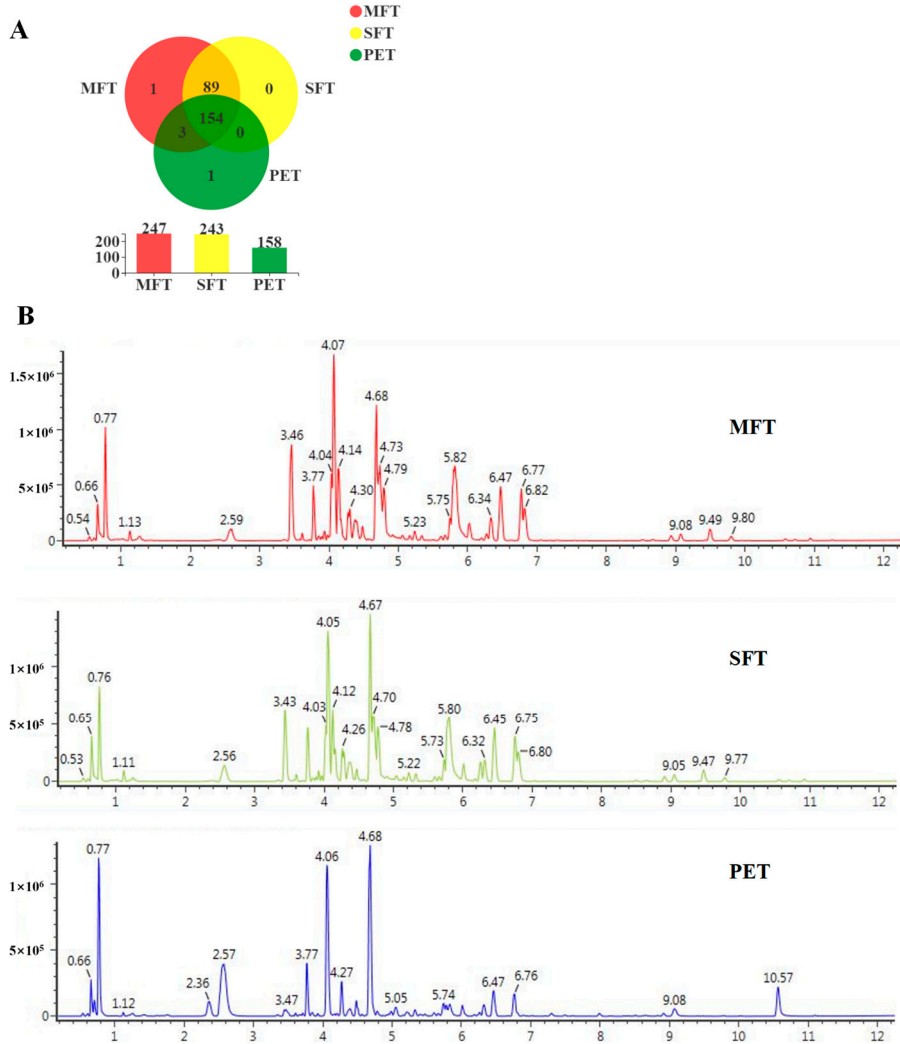

**Figure 4.** Metabolomic comparisons in different processing samples of Pu-erh tea. (**A**) Venn diagram of the identified metabolic compounds in different tea samples. (**B**) Total ion chromatograms in different samples of Pu-erh tea.

A total of 248 metabolites were subjected to multivariate analysis for an overview of the variance of the metabolites in different tea samples. HCA grouped the three samples into two clusters, with PET separated from MFT and SFT (Figure 5A). The PCA and OPLS-DA (Figure 5B,C) analyses showed a similar clustering, and the PET sample was visually different from the other two samples. The OPLS-DA models were well constructed and presented excellent predictive power (Figure 5D). Compared to the PET raw material, the SFT and MFT metabolite profiles significantly changed after the fermentation process. The concurrent changes in the fermentation tea samples revealed that microbiome changes were a critical factor in changing the metabolite profile.

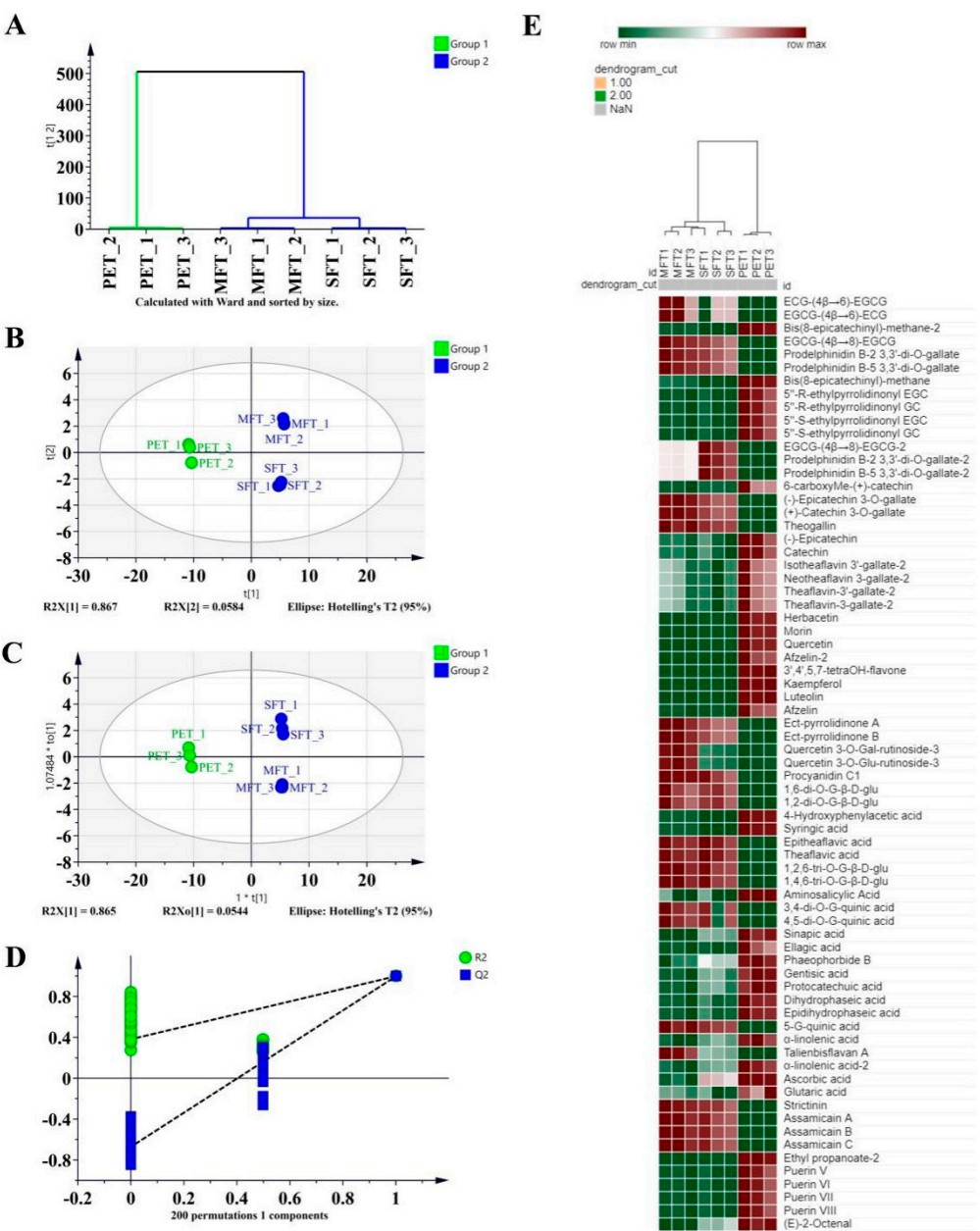

**Figure 5.** Multivariate analysis of metabolites from three tea samples. (**A**) Multivariate analysis of the three samples showing dendrogram plot of the individual tea samples. (**B**) PCA score plot. (**C**) OPLS-DA score plot. (**D**) Permutation plot of OPLS-DA. (**E**) Heat map of critical metabolites in three manufactured tea samples. Each column denotes a tea sample, and each row denotes a critical metabolite. The color-coded scale grading from green to brown corresponds to the level of critical metabolite shifting from low to high.

To explore the changes in metabolites during fermentation processing, a total of 71 critical metabolites were selected by OPLS-DA with good pairwise discriminations and a cut-off of variable importance projection (VIP) > 1.0. The VIP plots revealed the identified metabolites that contributed to the group separation (Figure S3). The OPLS-DA comparison between the PET and the other two groups revealed the differences in metabolites between the classes in each component. The critical metabolites were listed in a heat map for visual comparison (Figure 5E). Compared to PET, the key compounds epicatechin, catechin, herbacetin, morin, quercetin, afzelin, luteolin, gallate, syringic acid, sinapic acid, ellagic acid, ascorbic acid, glutaric acid, phaeophortide B, gentisic acid and puerin in MFT decreased significantly during the fermentation processes. The content of quinic acid, strictinin, assamicain and theogallin increased during spontaneous fermentation, but were further increased in microbial fermentation tea samples.

### 3.4. Critical Functional Components and Microorganisms Involved in Fermentation Processes

Metabolomics analysis is untargeted, and HPLC–MS is an analytical technique for the determination of small molecules. However, the nutritional and health-associated ingredients in Pu-erh tea are mostly macromolecules, such as polysaccharides, tea polyphenols and catechins. To understand the changes in critical functional components and their biological activities, tea polysaccharides, free amino acids, tea flavonoids, tea polyphenols and tea pigments were identified by biochemical methods.

Polysaccharides, as some of the main compounds contributing to tea flavor, are associated with a series of carboxylation reactions, such as intra-molecular dehydration, glycosylation and polymerization, producing a dark-brown soup color and caramel aroma under high-temperature and -humidity conditions. The content of tea polysaccharides remained stable after the spontaneous fermentation and microbial fermentation processes (Figure 6A).

The total content of flavonoids was markedly increased after the fermentation processes (Figure 6B). This may have been caused by the conversion of *L. plantarum*- and *S. cerevisiae*-derived flavone glycosides into flavonoids during microbial fermentation. Previous research reported that flavan-3-ols and procyanidin B3 contributed to the bitter and astringent tastes of a tea infusion [27]. The flavonoid changes may ameliorate the bitter and astringent tastes of Pu-erh tea.

Polyphenols, a health-promoting component of Pu-erh tea, were decomposed into chatein, chalignin and phenolic acid. Catechins are the main components, the proportion of which reaches 60–80% in tea polyphenols, and they contribute to the antioxidant activity of tea. After microbial fermentation, phloroglucinol and pyrogallol were generated and may have originated from the degradation of catechins, resulting in increased polyphenols after the manufacturing process (Figure 6C).

Amino acids are essential components contributing to the taste of Pu-erh tea. In total, 21 amino acids were determined by HPLC in the tea samples. The total amino acids were increased after microbial fermentation (Figure 6D).

The tea pigments theabrownin, theaflavin and thearubigin were analyzed with HPLC. The content of thearubigin, which contributes to the astringency and bitterness of tea, was significantly decreased after microbial fermentation (Figure 6E). On the contrary, theabrownin and theaflavin, which give a characteristic brown color to Pu-erh tea and play key roles in the quality of tea extracts, were increased after microbial fermentation.

### 3.5. Correlation between Critical Metabolites and Microorganisms

Multivariate analysis was used to study the correlation between critical metabolites and microorganisms after the fermentation of PET. Sixteen critical metabolites were selected for redundancy analysis (RDA) based on a variance inflation factor (VIF) < 10. Among the 16 critical metabolites, caffeine and EGC showed positive correlations with the fungal communities in the PET sample. Polyphenols and amino acids (Aa) showed positive correlations with most fungal communities in the MFT sample. Thearubigins

(TRs) and theobromins (TBs) showed negative correlations with the fungal communities in the SFT sample. TBs, theaflavins (TFs), flavonoids and epigallocatechin gallate (EGCG) showed negative correlations with most of the fungal communities (Figure 7A,B). *Aureobasidrurm leucispemi* and *Cutaneotrichosporon* played a positive role in producing caffeine and EGC. *Wallemia* and *S. cerevisiae* showed a correlation with polyphenols, Aa and EGCG in the microbial fermentation process. A positive correlation between the abundance of *Ceramothyrium carniolicum* and the content of polysaccharides, *Cryptococcus neoformans* and catechins was observed, as shown in Figure 7C. Our results clarified the effect of fungi on 16 critical metabolites by demonstrating that each metabolite corresponded to one or more fungal species.

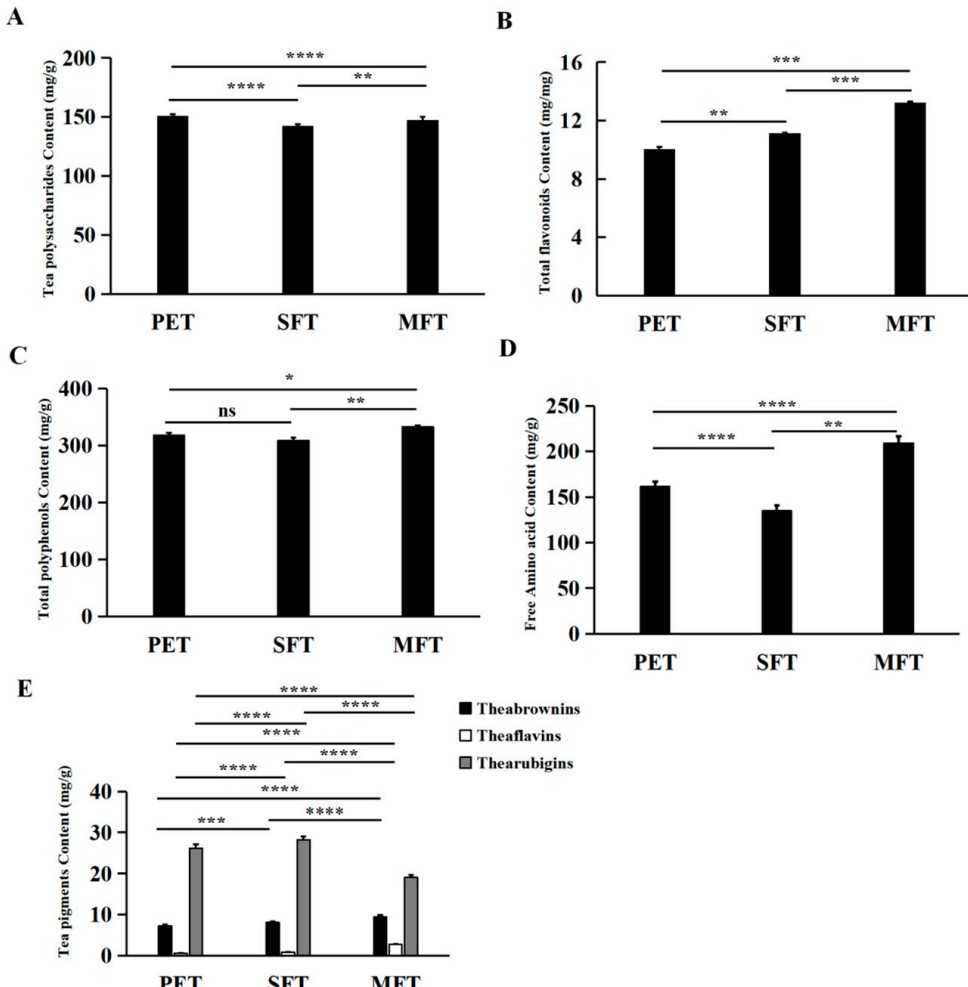

**Figure 6.** Functional components of tea samples measured by biochemical methods. (**A**) Changes in tea polysaccharides, (**B**) total flavonoids, (**C**) tea polyphenols, (**D**) free amino acids and (**E**) tea pigments were determined by HPLC; the data were obtained from three replicates. (Note: "ns", no significant differences; "*" $0.01 < p \leq 0.05$; "**" $0.001 < p \leq 0.01$; "***" $p \leq 0.001$; "****" $p \leq 0.0001$).

Bacteria affect the final concentrations of metabolites in tea. The active components are associated with the dominant bacteria in tea samples. Flavonoids, TBs, TFs, polyphenols and Aa exhibited negative correlations with most of the bacterial communities (Figure 8A). *Chloroplast* and *Pedobacter*, as the dominant bacteria in the PET sample, showed positive correlations with catechin and polysaccharides (Figure 8B). The dominant *Rosenbergiella* and *Enterobacterales* were involved in the consumption of caffeine and epigallocatechin (EGC) in the spontaneous fermentation process (Figure 8C).

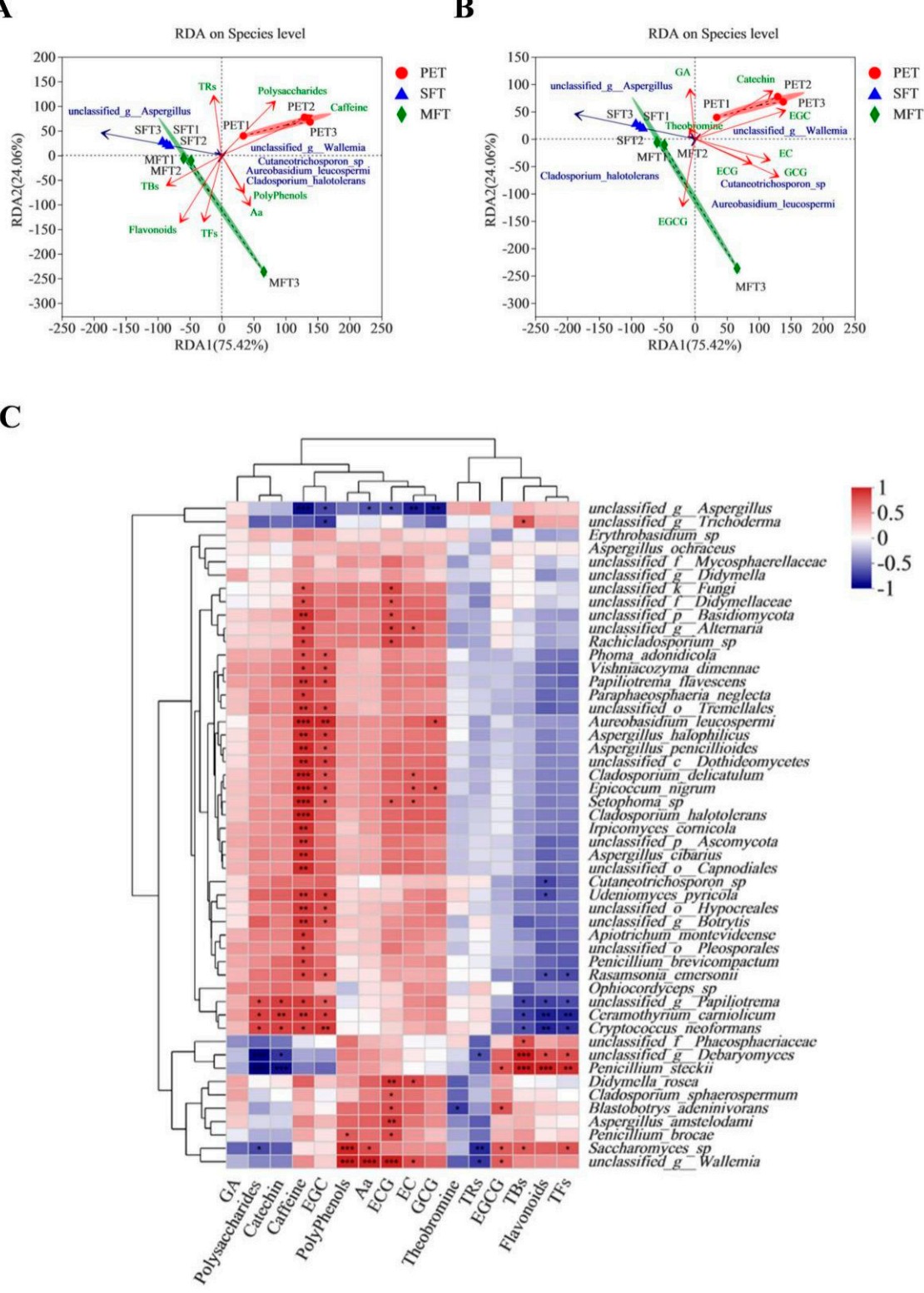

**Figure 7.** Correlations between fungal species and metabolites in different processed Pu-er tea samples. Redundancy analysis (RDA) shows the correlation between critical metabolites ((**A**): thearubigins, theobromins, flavonoids, theaflavins, amino acids, polyphenols, caffeine, polysaccharides; (**B**): gallic acid, theobromine, catechin, epigallocatechin, epicatechin, epicatechin gallate gallocatechin gallate, epigallocatechin gallate) and three Pu-er tea samples. (**C**) Heat map analysis of the correlation between critical metabolites and important fungal species. (Note: "*" $0.01 < p \leq 0.05$; "**"$0.001 < p \leq 0.01$; "***" $p \leq 0.001$).

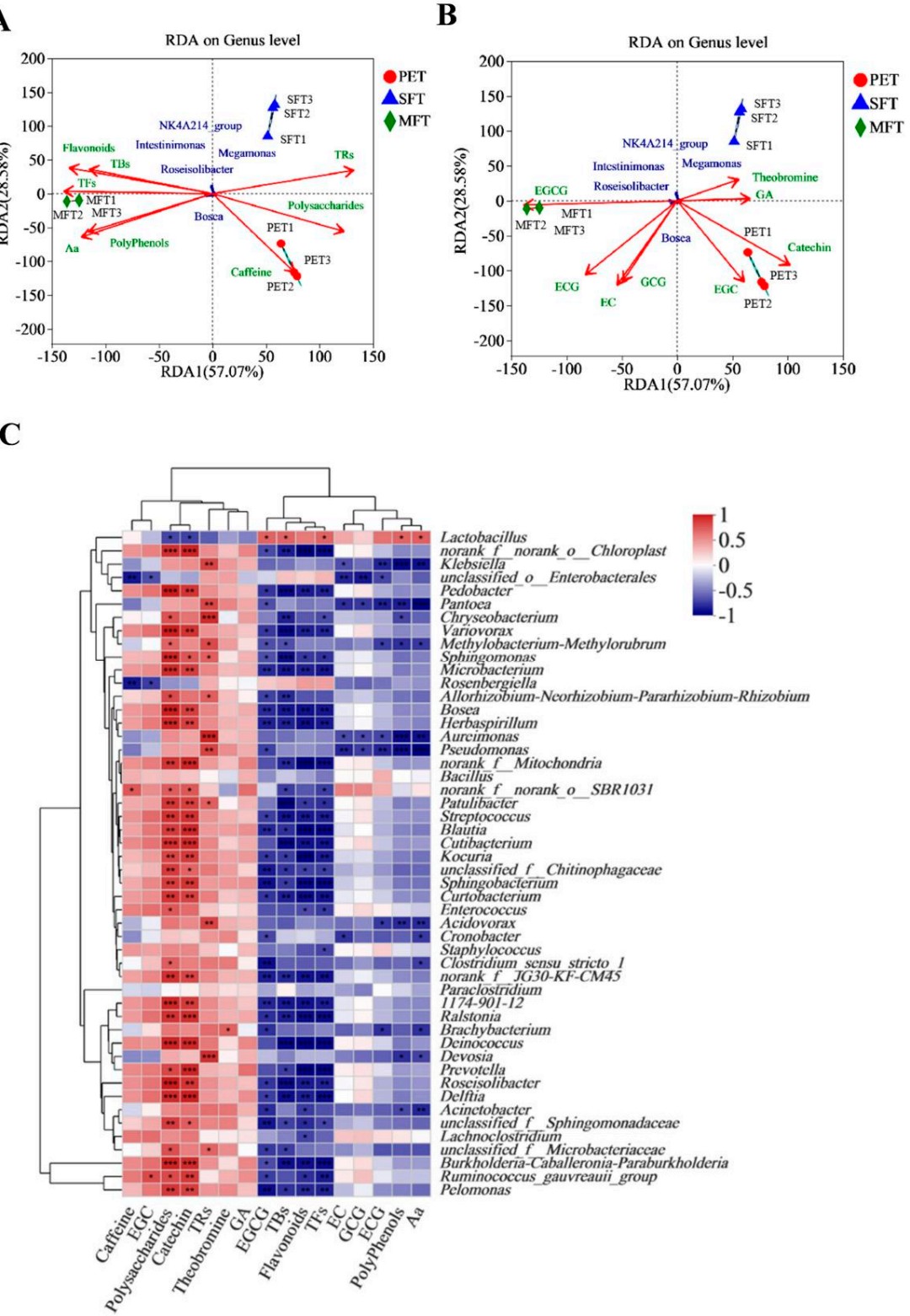

**Figure 8.** Correlations between bacteria and metabolites in different processed Pu-er tea samples. Redundancy analysis (RDA) shows the correlation between critical metabolites ((**A**): thearubigins, theobromins, flavonoids, theaflavins, amino acids, polyphenols, caffeine, polysaccharides; (**B**): gallic acid, theobromine, catechin, epigallocatechin, epicatechin, epicatechin gallate gallocatechin gallate, epigallocatechin gallate) and three Pu-erh tea samples. (**C**) Heat map analysis of the correlation between critical metabolites and important bacterial genera. (Note: "*" $0.01 < p \leq 0.05$; "**" $0.001 < p \leq 0.01$; "***" $p \leq 0.001$).

## 4. Discussion

*L. plantarum* and *S. cerevisiae* are critical microorganisms in the PET fermentation process, as they are the main contributors to enzyme secretion to change tea's functional components [12–14]. In our study, we compared the community modifications and metabolite changes in different manufactured PET samples by high-throughput Illumina MiSeq sequencing coupled with qPCR, HPLC–MS analysis, biochemical measurements and multivariate analysis. Fungal community analysis showed that *Aspergillus* was dominant in the three samples. *Aspergillus* was the predominant genus in the SFT sample, accounting for 99% of the total effective sequences. The *Aspergillus* in different species are responsible for producing amylase, acid protease, cellulase, pectinase and glucose oxidase, which contribute to degrading macromolecular organic matter and transforming substances [28–30]. *A. penicillioid* was the dominant species in the PET raw material, but was replaced by unclassified *Aspergillus* after fermentation processing. Based on the culture experiment and observation, we conjectured that the increase in unclassified *Aspergillus* was caused by *Eurotium cristatum*. In addition, the *S. cerevisiae* that were inoculated onto the tea leaves reproduced quickly in the MFT sample and competed with *A. penicillioides* for nutrients, resulting in suppressed unclassified *Aspergillus* growth during the fermentation.

At the genus level, *Chloroplast* was the most dominant bacteria in the PET raw material. The richness of *Klebsiella variicola*, unclassified *Enterobacterales* and unclassified *Pantoea* was increased during spontaneous fermentation processing. The richness of *L. plantarum* was the highest among the bacterial genera in the MFT sample. In addition, the diversity of bacteria was decreased, as the inoculated *L. plantarum* restrained other bacteria's growth.

In exploring the relationships between metabolism and microorganisms, a total of 248 metabolites were identified in the tea samples. Among them, 93 particular metabolites were from the MFT sample. These results indicated that microbial fermentation processing caused the microorganisms to undergo community structure changes and metabolic profile transformations. We predicted and analyzed the metabolic pathways of the microorganisms, and the relative abundance of pathways associated with carbohydrate transport and metabolism increased in MFT. This indicated that fungal metabolism was mainly involved in the hydrolysis of macromolecular compounds, while bacterial metabolism transferred small molecular compounds in the fermentation process. A total of 16 critical metabolites were screened for the metabolic variation study by LC–MS metabolomics and multivariate analysis. The results revealed that the critical metabolites were significantly different between the raw tea materials and fermented tea samples. The content of quinic acid, strictinin, assamicain and theogallin increased during spontaneous fermentation, but it was further increased in tea samples that underwent microbial fermentation, indicating that the biochemical profiles of SFT and MFT were influenced uniquely by the fermentation processes. Furthermore, the effect of bacteria on metabolites was stronger than that of fungi, which may have played an auxiliary role in substance transformation, which accelerated the transformation reaction and reduced the fermentation cycle.

The critical functional components and microorganisms involved in fermentation processes were studied to improve the flavor and quality of Pu-erh tea. From the microbiome results, *L. plantarum* and *S. cerevisiae* had varied effects on tea metabolites. Moreover, the metabolic analysis showed that polysaccharides, flavonoids and free amino acids were increased in the MFT sample. Although *S. cerevisiae* cannot secrete enzymes to break down polysaccharides, proteins and lipids during the fermentation process, we found that the metabolic activity of *S. cerevisiae* increased in the MFT. These changes probably resulted from *L. plantarum* secreting a variety of enzymes to promote *S. cerevisiae* growth, such as proteolytic and peptidelytic enzymes [31,32]. Meanwhile, the origin of enzymatic activity could be attributed to dead microbial cells or the intracellular fraction of plant cells, and some of the metabolites could also be products of spontaneous reactions. The polyphenols increased after manufacturing; this result was consistent with a report that galloylated catechins were hydrolyzed during microbial fermentation, concurrent with an increase in their hydrolytic products [33], which might be helpful in the bioactivity of MFT. The polysac-

charides remained stable after fermentation, which was likely because the water-soluble tea polysaccharides in PET are digested by hydrolytic enzymes from fungi and utilized as carbon sources for microorganisms in fermentation, and other polysacchrides composed of complexes with water-insoluble protein and lipid are released from insoluble complexes. Amino acids are important ingredients contributing to the taste of tea infusions. Previous research found that theanine and aspartic acid were positively correlated with the umami taste of a tea infusion [27]. The hydrolysis of extracellular enzymes from the secretion of microorganisms and consumption of microbes caused an increase in free amino acids. Protein was degraded into amino acids as a result of the hot/humid conditions and microbial catalysis. The changes in tea pigments (theabrownin, theaflavin and thearubigin) and their derivatives, flavonoids and flavonoid glycosides and simple phenols were consistent with the different dominant species in different tea samples [34]. Catechins are either oxidized to characteristic pigments or degraded into phenolic acids during fermentation processing. For example, catechin/EC/GC/EGC can be metabolized into diOH-phenylacetic acid via C-ring fission [33]. The growth of *L. plantarum* in MFT was closely related to the increase in EGCG, TBs, TFs, polyphenols and Aa in the fermentation process. In summary, the 16 critical metabolites had a strong correlation with bacterial genera, and the change in *L. plantarum* in the bacterial microbiome largely affected the metabolite transformation during fermentation.

This study reported the changes and correlations between the microbiome and metabolomics during the different manufacturing of Pu-erh tea samples. With a better understanding of the relationships between microorganisms and metabolites, it is possible to improve the compositions of bioactive compounds in PET through the exogenous inoculation of *L. plantarum* and *S. cerevisiae* to produce specific flavored PET and improve the quality of PET. Moreover, it is necessary to inoculate specific microorganisms and characterize the functional components from PET to further enhance the value of PET.

## 5. Conclusions

Investigating microbial fermentation in terms of producing beneficial bioactive compounds is a critical factor in Pu-erh tea manufacturing. To study the effects of the microbiome on the metabolite profile of fermented tea, *L. plantarum* and *S. cerevisiae*, two commonly studied microorganisms involved in fermentation, were inoculated onto tea leaves to increase the functional metabolites. Spontaneously fermented Pu-erh tea (no microbe pre-treatment) and raw material (non-fermented control leaves) were also studied as controls. To compare the three Pu-erh tea samples, the microbial community structure, correlation between microbiomes and metabolites and critical metabolite changes were characterized using metabolomics and microbiomics analyses. The spontaneously fermented tea displayed the lowest fungal diversity. More polyphenols, flavonoids and amino acids were identified in the *L. plantarum*- and *S. cerevisiae*-inoculated fermented tea (MFT). Compared to fungi, both the diversity and richness of bacteria were significantly decreased in the MFT sample. The conversion of metabolites in Pu-erh tea was mainly catalyzed by microbial enzymes secreted from microorganisms during the manufacturing process. In total, 71 critical metabolites were mostly responsible for the metabolic changes caused by the manufacturing process. Theabrownin, some novel phenolic acids and catechin derivatives formed, while the content of polysaccharides and tea pigments remained stable in MFT. These components are known to be responsible for the astringent and mellow tastes, as well as the brownish color and health benefits of Pu-erh tea.

In summary, our results advance the knowledge of the functions of *L. plantarum* and *S. cerevisiae* in the formation of the unique sensory characteristics of Pu-erh tea and reveal that the microbial composition is a critical factor in changing the tea's metabolic profile. These findings provide a new method to improve the quality and safety of Pu-erh tea.

**Supplementary Materials:** The following supporting information can be downloaded at: https://www.mdpi.com/article/10.3390/fermentation9110987/s1, Figure S1: High-throughput sequencing dilution Shannon curves for bacterial (A) and fungal (B); Figure S2: Bray-Curtis distance distribution of bacterial (A) and fungal (B) communities in different Pu-erh tea samples at the OTUs level; Figure S3: Variable importance of the projection (VIP > 1) plots of metabolites for distinguishing different sample groups using OPLS-DA model; Table S1: Control table of fungal metabolic activities (KEGG database); Table S2: Total metabolomics in all Pu-erh tea samples.

**Author Contributions:** Formal analysis, Z.S.; Investigation, Y.W., Y.L., Z.S., Y.Q., J.L. and X.W.; Writing—review & editing, J.B., J.L. and X.W.; Project administration, X.W.; Funding acquisition, Y.W., J.L. and X.W. All authors have read and agreed to the published version of the manuscript.

**Funding:** This work was supported by grants from the Shanghai Pujiang Talent Project (Grant No. 21PJ1411300), Shanxi Provincial Talent Engineering Project (Grant No. 2022TZRC01), Doctoral Scientific Research Foundation of Yulin University (2023GK030), Qin Chuangyuan High-Level Innovative and Entrepreneurial Talents Project (QCYRCXM-2023-055), National Key R&D Program of China (No. 2018YFC1604403) and the Fund of Shanghai Engineering Research Center of Plant Germplasm Resources (Grant No. 17DZ2252700).

**Institutional Review Board Statement:** Not applicable.

**Informed Consent Statement:** Not applicable.

**Data Availability Statement:** Data are contained within the article and Supplementary Materials.

**Acknowledgments:** The authors thank Jiang Bian for the help with the English writing.

**Conflicts of Interest:** The authors declare no conflict of interest.

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
