# Peer review of "Improving the Quality and Safety of Pu-erh Tea by Inoculation of Saccharomyces cerevisiae and Lactobacillus plantarum"

_fermentation, doi:10.3390/fermentation9110987_

Round 1

Reviewer 1 Report

Comments and Suggestions for Authors

The use of metabolomic analysis to characterize and to evaluate a fermented product is very interesting. Overall, the manuscript is well described and very readable. However, some points must be revised in order to improve the overall quality of manuscript.

The novel processing technology to improves the quality and safety of Pu-erh tea must be well-described in methodology section.

The metabolomic analysis must be described in the methodology

The Hierarchical cluster as well as principal component analysis must be included another subsection of material and methods section

Line 159. "MatabotAnalyst" should be "MetabotAnalyst"

Comments on the Quality of English Language

minor editing of English language is required

Author Response

Reviewer 1

The use of metabolomic analysis to characterize and to evaluate a fermented product is very interesting. Overall, the manuscript is well described and very readable. However, some points must be revised in order to improve the overall quality of manuscript.

The novel processing technology to improves the quality and safety of Pu-erh tea must be well-described in methodology section.

We thank the reviewer for critical evaluation of our manuscript and appreciating our work. We have made revisions to the Materials and Methods section to enhance the inoculated fermentation method in Line 115-119 “The Pu-erh tea raw materials (100 g) were mixed with distilled water (30 mL) to produce spontaneous fermentation tea (SFT). Li-mupirocin modified MRS medium containing Lactobacillus plantarum and Saccharomyces cerevisiae (10 mL, 1×108 Cells/mL) was sprayed on sun-dried tea leaves (100 g) and fermented at 37°C for 3 weeks, these samples were named microbial fermentation tea (MFT).”

The metabolomic analysis must be described in the methodology

This suggestion is well accepted. We added one paragraph to describe the metabolomics analysis method in Line 157-171.

The Hierarchical cluster as well as principal component analysis must be included another subsection of material and methods section

Thank you for your constructive feedback. We put this part to another section.

Line 159. "MatabotAnalyst" should be "MetabotAnalyst"

We apologize for the carelessness. The "MataboAnalyst” have been corrected to "MetaboAnalyst"

Reviewer 2 Report

Comments and Suggestions for Authors

Dear authors

 The submitted study provides valuable and complex information. I have only small comments and recommendations.

The manuscript can be acceptable after minor revisions.

 Comments and recommendations:

Fig 5.: „A color-coded scale grading from green to red“ - In my view, a color scale is graded from green to brown.

Fig. 6.: I recommend adding the results of statistical significance of differences above the columns.

References: The format of journal names is not unified (abbreviated vs. full, uppercase vs. lowercase initial letters).

Author Response

Reviewer 2

The submitted study provides valuable and complex information. I have only small comments and recommendations.

The manuscript can be acceptable after minor revisions.

 Comments and recommendations:

Fig 5.: „A color-coded scale grading from green to red“ - In my view, a color scale is graded from green to brown.

Thanks for your suggestion, it is well accepted and we changed it to brown in Line 339.

Fig. 6.: I recommend adding the results of statistical significance of differences above the columns.

We very appreciate the reviewer’s concerns. We added the statistical significance to difference above the columns in figure 6.

References: The format of journal names is not unified (abbreviated vs. full, uppercase vs. lowercase initial letters).

We have checked all the references and formatted them strictly according to the Guide for Authors. Especially, journal names have been abbreviated.

Reviewer 3 Report

Comments and Suggestions for Authors

The manuscript “A novel processing technology improves the quality and safety of Pu-erh tea through the combined exogenous inoculation of Saccharomyces cerevisiae and Lactobacillus plantarum (Fermentation journal_2692327) compares the raw material tea, spontaneous fermentation tea and microbial fermentation tea, which were analysed to investigate microbiome composition changes and the correlation between microbiome and metabolites. Particularly, the authors have reported that the fermented tea with inoculated microorganisms showed more changes in metabolite composition. The results presented provide insight into the tea fermentative behaviour using S. cerevisiae and malolactic bacteria in order to obtain teas with complex aroma and healthy composition.

The manuscript was written in a manner that does not conforms to the standard scientific method. A reasonable number of references were cited as evidence of substantive background. The methods and experimental procedures were described in sufficient detail to enable future researchers to follow up on aspects of the authors’ work. The results were presented in a clear manner using graphical and tabular means with appropriate statistical analysis, but the results section is mixed up with the discussion section. In other ways, the readers will find twice that kind of comment. That problem is present many times in the results section. On the other hand, in the discussion, the authors are referring only to two papers, so what are they discussing? A few conclusions were drawn, and few suggested areas for future experimentation were outlined. Although the results in this manuscript are quite interesting, they should be presented correctly.

Major points:

This reviewer suggests to the authors that they present the secondary metabolites separately from the primary, which could help to distinguish the real variations mediated by the inoculated and non-inoculated fermentations.

Minor Points:

P1 L2: The title is too long; this reviewer suggests a shorter version as “Improving quality and safety of Pu-erh tea by inoculation of Saccharomyces cerevisiae and Lactobacillus plantarum

P1 L20: The authors are using two microbial fermentations: inoculated vs. spontaneous.

P1 L31: The authors should not forget the reproducibility of the characteristics of the product by using microbial starter fermenters.

P3 L115: For this reviewer, 37ºC is too high and could enhance the loss of volatile compounds. Why do the authors use this fermentation temperature, which is not the optimum for these microorganisms?

P3 L115: The authors must mention which are the aerobic conditions during fermentation: anaerobic strict, oxygen restriction, etc. That condition could help the reader understand the differences in the composition of the evaluated microbiomes.

P3 L116: This reviewer assumes that the fermentations were done in dark conditions, is that right?

P4 L157: The authors must mention that the samples are compounded by vegetable material and the microorganisms. That will help to understand metabolic results.

P4 L179: These results show "Microbiome characterization of the different Pu-erh teas",

please join punts 3.1 and 3.2.

P5 L210: The significant of PET, SFT and MFT must be enclosed into the figure legend.

P6 L219: Tis reviewer does not understand sterilized concept here, because it was not mention in the section M&M.

P6 L220: Change Aspergillus_penicillioides by A. Penicillioides. On the other hand, this comment is valid for all the organism names in the document. The second time that the organism is mentioned, it could be used in its abbreviated version.

P7 L248: This affirmation is not true, SFT from Figure 2 does not show that. What is the means of growth in this sentence?

P7 L258: That concept was already mentioned at the summary and at the introduction, please remove it from here.

P8 L281: This section is Results, so the hypothetical scenarios that could justify a result must be in the "Discussion" section. In other ways, the readers will find twice that kind of comment. That problem is present many times in this section.

P9 L301: The authors must mention that the number of different compounds between MFT and SFT is represented by 1 compound in the MFT sample. The authors must write up what they observed in the figures. This comment is valid for this section all.

P11 L324: The authors used only one time point for the fermentations; because they do not have data at different time points of the fermentation process, this assertion is not true.

P12 L371: What are those neurotransmitters in microorganisms? Please, the authors must modify this document in accordance with the basic biological concepts.

P13 L375: This reviewer does not agree with this concept. Free AAs are always used by the microorganisms, and the MFT shows higher values than the SFT, which is contradictory. The sample with fewer cellulolytic enzymes has a lower capacity to break the cell wall of plant cells. From where are coming these AA?

P13 L393: This reviewer does not agree with this comment because the polyphenols are not secondary products of S. cerevisiae or Wallemia sp.

P14 L407: This affirmation is not true. The bacteria could affect the final concentration of metabolites.

P16 L428: This sentence is not clear. On the other hand, the authors insisted on the activity of secreted enzymes, but they have not evaluated that. The origin of enzymatic activities could be provided by dead microbial cells or by the intracellular fraction of plant cells. The author evaluated the metabolites and found that some of them could also be products of spontaneous reactions.

P16 L434: Please, only mention the specific enzymes from Aspergillus and their putative role in this specific fermentation.

End of review

Comments on the Quality of English Language

Some words and concepts used by the authors are not appropriate for microorganisms.

Author Response

Reviewer 3

The manuscript “A novel processing technology improves the quality and safety of Pu-erh tea through the combined exogenous inoculation of Saccharomyces cerevisiae and Lactobacillus plantarum (Fermentation journal_2692327) compares the raw material tea, spontaneous fermentation tea and microbial fermentation tea, which were analysed to investigate microbiome composition changes and the correlation between microbiome and metabolites. Particularly, the authors have reported that the fermented tea with inoculated microorganisms showed more changes in metabolite composition. The results presented provide insight into the tea fermentative behaviour using S. cerevisiae and malolactic bacteria in order to obtain teas with complex aroma and healthy composition.

We thank the reviewer for detailed evaluation of our manuscript.

The manuscript was written in a manner that does not conforms to the standard scientific method. A reasonable number of references were cited as evidence of substantive background. The methods and experimental procedures were described in sufficient detail to enable future researchers to follow up on aspects of the authors’ work. The results were presented in a clear manner using graphical and tabular means with appropriate statistical analysis, but the results section is mixed up with the discussion section. In other ways, the readers will find twice that kind of comment. That problem is present many times in the results section. On the other hand, in the discussion, the authors are referring only to two papers, so what are they discussing? A few conclusions were drawn, and few suggested areas for future experimentation were outlined. Although the results in this manuscript are quite interesting, they should be presented correctly.

We appreciate the reviewer careful consideration of these issues. We have revised the whole manuscript carefully and tried to write in a manner to fit this journal. We have checked all the references and cited more reference to evidence the substantive back ground. The material and method section have been re-written in sufficient detail. As reviewer suggestion, these discussions in results section have been removed. In addition, we have checked the result section several times. We believe that the result is now acceptable for the review’s requirements. We have gone through the discussion section and added some important references to support our conclusions. The conclusions have been revised to present the interesting of this work.

Major points:

This reviewer suggests to the authors that they present the secondary metabolites separately from the primary, which could help to distinguish the real variations mediated by the inoculated and non-inoculated fermentations.

Thanks for your constructive feedback. Of course, if we present the secondary metabolites separately from the primary, which could help to distinguish the real variations mediated by the inoculated and non-inoculated fermentations. However, in our article, we have compared the differences of metabolites between fermentation and non fermentation tea samples. Even if the metabolites are not isolated, it is directly to see the differences of the metabolite in inoculated and non-inoculated fermentation tea samples.

Minor Points:

P1 L2: The title is too long; this reviewer suggests a shorter version as “Improving quality and safety of Pu-erh tea by inoculation of Saccharomyces cerevisiae and Lactobacillus plantarum“

As the reviewer suggestion, the title have been changed to “Improving the quality and safety of Pu-erh tea by inoculation of Saccharomyces cerevisiae and Lactobacillus plantarum”

P1 L20: The authors are using two microbial fermentations: inoculated vs. spontaneous.

These two sentences in the introduction have been combined and modified to “Traditional Pu-erh tea is fermented spontaneously using raw materials in hot and humid conditions without a seeding starter strain, this fermentation....”

P1 L31: The authors should not forget the reproducibility of the characteristics of the product by using microbial starter fermenters.

This is a good suggestion, we will explore the reproducibility of the characteristics of the product after using saccharomyces cerevisiae and lactobacillus plantarum as starter in the future research.

P3 L115: For this reviewer, 37℃ is too high and could enhance the loss of volatile compounds. Why do the authors use this fermentation temperature, which is not the optimum for these microorganisms?

The temperature profile revealed 37°C as optimum for the growth of lactobacillus plantarum, as well as saccharomyces cerevisiae. Moreover, pile-fermentation is usually used for manufacturing Pu-erh tea in the Yunnan tea industry. During this process, the temperature could reach 35-40°C inside of tea. Overall, 37°C was selected for fermentation temperature.

P3 L115: The authors must mention which are the aerobic conditions during fermentation: anaerobic strict, oxygen restriction, etc. That condition could help the reader understand the differences in the composition of the evaluated microbiomes.

Thank you for pointing this out. The details of condition have been added in line 118.

P3 L116: This reviewer assumes that the fermentations were done in dark conditions, is that right?

The fermentation condition not required to avoid light, so we did not mention this fermentation done in dark condition.

P4 L157: The authors must mention that the samples are compounded by vegetable material and the microorganisms. That will help to understand metabolic results.

We are thankful for the provided feedback. To provide further clarification, we have enhanced the describe in Line 158-159.

P4 L179: These results show "Microbiome characterization of the different Pu-erh teas",

please join punts 3.1 and 3.2.

We have now added the different Pu-erh teas in Line 203.

P5 L210: The significant of PET, SFT and MFT must be enclosed into the figure legend.

As requested by Reviewer, we have added significant of PET, SFT and MFT in Figure 1B.

P6 L219: Tis reviewer does not understand sterilized concept here, because it was not mention in the section M&M.

Thank you for insightful suggestion. We have deleted this word “sterilized” in Line 24.

P6 L220: Change Aspergillus_penicillioides by A. Penicillioides. On the other hand, this comment is valid for all the organism names in the document. The second time that the organism is mentioned, it could be used in its abbreviated version.

Thank you for your suggestion. We have revised all the organism names in this manuscript.

P7 L248: This affirmation is not true, SFT from Figure 2 does not show that. What is the means of growth in this sentence?

This sentence has been changed to “Klebsiella variicola and unclassified Enterobacterales were increased in the SFT sample.”

P7 L258: That concept was already mentioned at the summary and at the introduction, please remove it from here.

As requested by Reviewer, this sentence has been removed.

P8 L281: This section is Results, so the hypothetical scenarios that could justify a result must be in the "Discussion" section. In other ways, the readers will find twice that kind of comment. That problem is present many times in this section.

Thank you for your constructive feedback. We have removed “During fermentation process, hundreds of metabolic activities closely relate with fungal communities which contribute to the formation of special flavors of Pu-erh tea. ”

P9 L301: The authors must mention that the number of different compounds between MFT and SFT is represented by 1 compound in the MFT sample. The authors must write up what they observed in the figures. This comment is valid for this section all.

Thank you for your suggestion. We have revised this sentence to “Among the remaining 94 metabolites, 93 metabolites were from the MFT sample and 1 metabolite from PET sample. Moerover, 89 metabolites were specific to fermentation tea samples.”

P11 L324: The authors used only one time point for the fermentations; because they do not have data at different time points of the fermentation process, this assertion is not true.

As recommended, we have revised the description in Line 337.

P12 L371: What are those neurotransmitters in microorganisms? Please, the authors must modify this document in accordance with the basic biological concepts.

We have removed the incorrect statement.

P13 L375: This reviewer does not agree with this concept. Free AAs are always used by the microorganisms, and the MFT shows higher values than the SFT, which is contradictory. The sample with fewer cellulolytic enzymes has a lower capacity to break the cell wall of plant cells. From where are coming these AA?

We would like to thank you for your insightful opinions. Our results showed that the MFT shows higher values of free AAs than that of SFT. We speculated that most of the macromolecules in the tea matrix were hydrolyzed or converted into small molecules during the fermentation process due to abundant fungi appearing.

P13 L393: This reviewer does not agree with this comment because the polyphenols are not secondary products of S. cerevisiae or Wallemia sp.

Thank you for your insightful comment. This describe have been changed to “Wallemia and S. cerevisiae have the correlation with polyphenols”.

P14 L407: This affirmation is not true. The bacteria could affect the final concentration of metabolites.

We agree and have revised the incorrect statement “Bacteria affect the metabolism of endogenous compounds in tea” to “Bacteria affect the final concentration of metabolites in tea.” Thank you.

P16 L428: This sentence is not clear. On the other hand, the authors insisted on the activity of secreted enzymes, but they have not evaluated that. The origin of enzymatic activities could be provided by dead microbial cells or by the intracellular fraction of plant cells. The author evaluated the metabolites and found that some of them could also be products of spontaneous reactions.

We very appreciate the reviewer’s concerns. We have added the reviewer’s comments in our discussion. Meanwhile, the origin of enzymatic activities could be provided by dead microbial cells or by the intracellular fraction of plant cells and some of metabolites could also be products of spontaneous reactions.

P16 L434: Please, only mention the specific enzymes from Aspergillus and their putative role in this specific fermentation.

Thank you for your feedback. We revised our statement “The Aspergillus in different species are provided for producing of amylase, acid protease, cellulase, pectinase, glucose oxidase, which could apply for degrading macromolecular organic matter and transforming substances ”.

Round 2

Reviewer 3 Report

Comments and Suggestions for Authors

This reviewer does not have further comments.

Author Response

Thanks for your review.